# Schizotypy, Lifestyle Behaviors, and Health Indicators in a Young Adult Sample

**DOI:** 10.3390/bs11120179

**Published:** 2021-12-16

**Authors:** Thomas J. Dinzeo, Uma Thayasivam

**Affiliations:** 1Department of Psychology, Rowan University, Glassboro, NJ 08028, USA; 2Department of Mathematics, Rowan University, Glassboro, NJ 08028, USA; thayasivam@rowan.edu

**Keywords:** schizotypy, schizotypal personality, psychosis-risk, health, lifestyle, somatic symptoms, sleep, nicotine, alcohol and substance use, BMI

## Abstract

Problematic lifestyle behaviors and high rates of physical illness are well documented in people with schizophrenia, contributing to premature mortality. Yet, there is a notable absence of research examining general lifestyle and health issues in participants at risk for psychosis. This form of research may help identify concerns that exist during prodromal periods related to future outcomes. Accordingly, the current study examined lifestyle and health in a nonclinical sample of 530 young adults with varying levels of schizotypy. Increasing symptom severity was associated with greater somatic symptoms and poorer sleep quality across positive, negative, and disorganized domains. Elevated negative and disorganized symptoms were associated with significantly reduced health-related quality of life, while evidence for reduced engagement in health behaviors was largely limited to those with elevated negative schizotypy. No relationships emerged between symptom presentation/severity and body mass index or substance use, although zero-order correlations suggested an association between disorganized schizotypy and nicotine use. The pattern of relationships in the current study was consistent with findings from the ultra-high risk and clinical literature suggesting that lifestyle and health concerns may exist on a continuum with psychosis. Future research should seek to clarify if these patterns are associated with long-term physical or mental health outcomes.

## 1. Introduction

Individuals diagnosed with schizophrenia tend to have higher rates of obesity, physical illness, and premature mortality when compared to the general public [1,2,3,4,5]. Overrepresented chronic illnesses include hypertension, cardiovascular disease, diabetes, and respiratory illness [6,7,8,9]. Functional disruptions caused by symptoms and/or the side effects of antipsychotic medications may contribute to the deterioration of physical health. These issues are compounded by lifestyle behaviors including unhealthy patterns of eating, low levels of physical activity, and high rates of smoking [10,11,12,13,14,15]. However, there is accumulating evidence suggesting that a number of physical health issues (e.g., physical illness and somatic symptoms) [16,17] and lifestyle concerns [18,19,20,21,22] exist prior to the development of clinically significant symptoms or prolonged treatments. These findings are noteworthy since they suggest the need to evaluate physical health parameters, and the presence/absence of health behaviors, in those at-risk for psychosis. Unfortunately, information relating to lifestyle and physical health is inconsistently collected in those at-risk for psychosis contributing to considerable gaps in our collective knowledge [23]. There is evidence in the literature for reduced physical activity and higher caloric consumption when comparisons are made to control participants [19,22]. At-risk participants also tend to have higher nicotine, alcohol, and cannabis use [22,24,25]. Problematic use of cannabis may potentiate psychotic symptom expression [26,27,28,29,30,31]. Existing biological/genetics data suggest that the use of these substances may represent the influence of an underlying genetic/biological process related to the pathophysiology for schizophrenia [32].

In addition to specific lifestyle behaviors, there is also evidence suggesting that individuals with an increased risk for schizophrenia may experience a higher prevalence of physical health concerns and sleep disturbance. For example, adolescents meeting the criteria for schizotypal personality disorder reported greater health-related concerns (illness/injury or inability to obtain treatment) when compared to controls or participants with other personality disorder diagnoses [17]. Similar findings were reported in a large sample of undergraduate students, where levels of schizotypy were negatively correlated with health-quality of life [33,34,35]. These reported physical ailments and health dissatisfaction warrant further investigation since they may be related to immune response/inflammation [36] or other potentially etiologically relevant processes. Poorer physical health is generally associated with sleep disruptions and greater mental health symptomatology [37,38,39]. Notably, increased sleep latency, poor sleep continuity, and disrupted sleep architecture are commonly reported in those at-risk for psychosis [40,41,42,43,44,45]. Sleep disruption may indicate important clinical changes foreshadowing the emergence of more severe symptoms or behaviors [46,47] while depleting the cognitive and emotional resources that are generally protective [43]. Yet, the existing literature examining sleep disturbance and risk for psychosis is relatively circumscribed, focusing on issues of sleep and clinical symptoms without integration of other lifestyle/health indicators.

Collectively, these findings suggest that health-related concerns may exist prior to diagnosis/treatment of psychosis and may be relevant to understanding prospective mental and physical health outcomes. The majority of existing research on this topic involves ultra-high risk (UHR) participants who generally have some degree of ongoing clinical care (e.g., use of antipsychotic medications). Thus, research examining risk variables in non-clinical samples can provide information about the presence/development of health-related issues while reducing confounds associated with clinical settings [48].

The purpose of the current study was to examine the relationship between levels of schizotypy, lifestyle, and health indicators in a non-clinical sample of young adults. Based on the associated literature, we anticipated that higher levels of schizotypy would be associated with poorer health behaviors (e.g., reduced activity, increased smoking and substance use, poorer sleep quality) and indications of poor physical health (physical symptoms; health-related quality of life). Evidence for differential health and lifestyle patterns could have potential implications for existing risk-assessment protocols, etiological models of psychosis, as well as the consideration of preventative care strategies for those at-risk.

## 2. Materials and Methods

### 2.1. Participants

Participants were recruited from a midsized university in the northeastern portion of the US. Students had the option to participate in research or non-research activities to fulfill a course requirement. The study was approved by the university IRB and was conducted in accordance with the Declaration of Helsinki. All participants, who were required to be 18 years or older, received detailed information about the nature of the study and provided consent prior to participation. While 550 enrolled in the study, the final sample was reduced to 530 after removing participants with incomplete data. Participants were provided a link allowing them to complete questionnaires online. The mean age of the sample was 20.42 years, 51.4% were male, with 74.6% Caucasian, 12.1% African American, 10.2% Latino/a, 2.6% Asian-Pacific, and 6% Native American participants.

### 2.2. Measures

**Schizotypy:** Participants completed the Schizotypal Personality Questionnaire-Brief Revised (*SPQ-BR*) [49]. This measure is composed of 32 statements that are rated on a Likert-type scale ranging from 1 (Strongly Disagree) to 5 (Strongly Agree) that combine into a total score. This measure can also be used to evaluate three empirically derived subdomains including (1) *Interpersonal* (negative) schizotypy; higher scores reflecting discomfort in social situations, difficulty expressing emotions or feeling close to others, (2) *Disorganized* schizotypy; higher scores reflecting odd speech or eccentric behavior, and (3) *Cognitive-perceptual* (positive) schizotypy; higher scores on this subscale which illustrates odd perceptual experiences, magical thinking, and suspicious beliefs. Each subscale has demonstrated good internal consistency across multiple large non-clinical samples (Cronbach’s α = 0.80–0.90) [50].

**Lifestyle behaviors:** The 42-item Lifestyle and Habits Questionnaire-Brief (*LHQ-B*) [51] was administered as a broad indicator of 8 lifestyle domains including physical health and exercise, psychological health, substance abuse, nutrition, environmental concern, social concern, accident prevention, and sense of purpose. Statements from each domain (e.g., nutrition: “*I eat five or more servings of fruits and vegetables daily*”) are rated on a scale from 1 (Strongly Disagree) to 5 (Strongly Agree). The internal consistency of the LHQ-B domains ranges from fair to excellent (0.65 to 0.91) in college samples [51]. The seven-item International Physical Activity Questionnaire (IPAQ)-short version [52] was administered to establish the amount of physical activity (i.e., vigorous, moderate, walking) during the 7 day period prior to the study. The duration of activities (in minutes) was multiplied by the associated MET units (Metabolic Equivalent of a Task) where *walking* = 3.3 METs, moderate activity = 4.0 METs, and vigorous activity = 8.0 METs. The resulting scores were then summed to provide an estimate of overall energy expenditure with scores below 600 MET minutes per week indicating low levels of physical activity, scores between 600 and 2999 indicating moderate levels of activity, and scores above 3000 indicating high levels of activity. This measure was developed for individuals 15–69 years of age with demonstrated psychometric properties [53]. The Pittsburgh Sleep Quality Inventory (PSQI) [54] is a 19-item measure with seven subscales representing different elements of sleep quality (e.g., sleep latency; sleep disturbance). These subscales are summed to yield a total score ranging from 0 to 21, with higher total scores indicating poorer sleep quality. Scores above 6 are likely to indicate the presence of insomnia disorder in US college students [55]. Nicotine use was assessed by asking the number of times per day (on average) that participants smoked cigarettes or used other nicotine products (chewing tobacco, cigars, etc.). Alcohol use was evaluated through an estimate of alcoholic products consumed in a typical week. To measure illicit substance use, a series of screening questions were used to evaluate use during the past month. If use was endorsed, then participants were prompted with specific questions regarding frequency and amount of use. For each category of substances (marijuana, cocaine, amphetamines, sedatives, opioids, hallucinogens) we provided other common names to facilitate reporting (e.g., amphetamine, methamphetamine, meth, speed, Adderall).

**Physical health indicators:** The 33-item Cohen-Hoberman Inventory of Physical Symptoms (CHIPS) [56] was administered to assess the presence of somatic symptoms rated from 0 (not bothersome) to 4 (extremely bothersome) during the previous 2-week period. Examples include symptoms such as stomach pain, headache, muscle tension or soreness, indigestion, shortness of breath, numbness or tingling in extremities, nose bleeds, and blurred vision. Scores were summed to create the CHIPS total score, with higher scores indicating a greater range and severity of physical symptoms. In addition, eight CHIPS symptom clusters [57] were composed across the following domains (1) *sympathetic/cardiac* (e.g., racing heart, shortness of breath, tingling in body), (2) *muscular* (e.g., pain, cramps, soreness, tension), (3) *metabolic* (e.g., low energy, poor appetite, weight change), (4) *gastrointestinal* (e.g., indigestion, constipation, nausea), (5) *vasovagal* (e.g., fainting or dizziness), (6) *cold/flu* (e.g., cough, stuffy head/nose), (7) *headache* and (8) *minor hemorrhagic* (nose bleeds, bruising). Scores were summed and then averaged by dividing by the number of items representing each cluster. These groupings demonstrate adequate construct and discriminant validity when compared to other established somatic indices [57]. The Quality of Life Inventory (QOLI) health scale [58] was administered to quantify participants’ overall satisfaction with their physical fitness and the absence of illness, pain, and disability [59]. Items were rated on scales representing satisfaction with health ranging from −3 (very dissatisfied) to +3 (very satisfied) and the importance of health on participants’ overall happiness ranging from 0 (not important) to 2 (extremely important). An overall score was derived by multiplying the two scales together, which results in a score ranging from −6 to 6 with higher scores indicating better overall wellbeing. Body Mass Index (BMI) was calculated using a formula provided by the Center for Disease Control and Prevention [60] where weight in pounds is divided by height in inches squared; this product is then multiplied by 703. The resulting score can be used as an indicator of body fat, with higher scores generally representing greater body mass [61].

### 2.3. Data Preparation and Analyses

Prior to the analyses, the data were examined for potential confounds or violations of normality. All variable distributions met the normal assumptions with the exception of the substance use variables where the majority of individuals reported no, or little, use. There were low endorsement rates (<1%) for most substances (excluding them from analyses) except for alcohol (72%), marijuana (24%), and nicotine (20%). Pearson’s bivariate correlations were used to examine the strength and direction of study variables with levels of schizotypy in continuous form. Spearman’s rank order bivariate correlations (non-parametric) were applied to alcohol, marijuana, and nicotine since they contained excessive skew (<2). We created three symptom score categories (high, intermediate, low) based on cut-offs used in previous research [62]. SPQ-BR scores falling 1.65 standard deviations above the sample mean (<95 percentile) were designated as *high* (i.e., *psychometrically defined schizotypy*). *Low* schizotypy was designated by scores falling below the mean for the SPQ-BR, while *intermediate* scores fell between the low and high score cutoffs. A three-way MANOVA was conducted to examine gender, ethnicity, and levels of schizotypy on lifestyle and health variables (i.e., level of physical activity, BMI, lifestyle subscales, physical symptoms, sleep quality, and use of nicotine/alcohol/drugs). All analyses were conducted using SPSS version 27.

## 3. Results

Pearson’s bivariate correlations were conducted in order to examine the general direction and magnitude of associations (zero-order) between all study variables (Table 1) with a Bonferroni-corrected level of significance set at *p* = 0.002 (i.e., *p* = 0.05/18).

Prior to MANOVA analyses, we examined the dataset for multivariate normality of the independent variable (levels of schizotypy) utilizing the Shapiro–Wilk test and utilizing Box’s Test to examine the homogeneity of covariances across groups. Due to evidence for heterogeneity in the DVs covariances, we used the Pillai’s Trace test since it is not highly linked to assumptions about the normality of data distribution [63]. Pillai’s trace values for each subdomain were: *positive,* 0.198, *F* (34, 1024) = 3.306, *p* < 0.001; partial eta squared = 0.099 with power to detect the effect = 1.0; *negative*, 0.234, *F* (34, 1024) = 4.000, *p* < 0.001; partial eta squared = 0.117 with power to detect the effect = 1.0; *disorganized,* 0.153, *F* (34, 1024) = 2.495, *p* < 0.001; partial eta squared = 0.077 with power to detect the effect = 1.0. The MANOVA main effects were significant, providing support for the basic premise of the study and supporting additional analyses: *positive, F* (17, 511) = 1077.82, *p* < 0.001), *negative, F* (17, 511) = 1057.77, *p* < 0.001), disorganized, *F* (17, 511) = 921.14, *p* < 0.001). The univariate main effects for each lifestyle and health indicator using a Bonferroni corrected level of significance (i.e., *p* = 0.05/16 = *p* = 0.003) can be viewed in Table 2, Table 3 and Table 4. Partial Eta Squares (PES) indicated relatively large effect sizes, with the Observed Powers (listed as Power) above 91% (0.91), for all significant variables suggesting sample size was adequate. Significant effects for gender and ethnicity were indicated; *F* (17, 506) = 8.483, *p <* 0.001 and *F* (68, 2036) = 2.110, *p <* 0.001, respectively, with females reporting more severe somatic symptoms (CHIPS), less substance use, better nutritional habits, greater social concern, more accident prevention awareness, and greater sense of purpose while males reported more engagement in health and exercise behaviors (LHQ-B). The post-hoc column in Table 2, Table 3 and Table 4 provides an indication of significant results between groups noted by a greater-than (>) or less-than (<) symbol depending on the direction of the difference and how measures are scored.

Post-hoc analyses suggested that individuals in the high (H) schizotypy category in all three symptom domains (i.e., positive, negative, and disorganized) reported more severe somatic symptoms, lower psychological health (LHQ-B), and more sleep-related difficulties (PSQI) when compared to individuals in the low (L) schizotypy groups. In each instance, the intermediate (I) schizotypy group reported more severe somatic symptoms, lower psychological health (LHQ-B), and poorer sleep quality (PSQI) when compared to the low schizotypy group. In addition, the high negative schizotypy group reported reduced engagement in health behaviors (LHQ-B), reduced sense of purpose (LHQ-B) poorer health-related quality of life (QOLI) when compared to the low schizotypy group, with significant differences also found between intermediate vs. low groups (Table 3). Reduced engagement in health behaviors was also found in participants with high vs. low disorganized schizotypy, but this was considered a statistical trend (*p* = 0.008). Significant main effects were found for greater nicotine use and reduced engagement in health behaviors in participants with increasing levels of disorganized schizotypy (Table 4). However, the post-hoc comparison suggested that the group differences in nicotine use (*p* = 0.02) and health behaviors (*p* = 0.008) were not statistically significant in the context of the Bonferroni correction requiring *p* < 0.003 and are therefore listed as a trend.

ANOVAs were used to examine group differences across specific somatic symptoms and sleep quality to obtain a clearer picture of what specific components of these variables were related to levels of schizotypy. Bonferroni-corrected level of significance set at *p =* 0.007 (i.e., *p =* 0.05/7). The results suggested that stepwise increases of somatic symptoms were present across all somatic symptom categories except for cold/flu symptoms (Figure 1).

Similarly, group differences were examined across the seven composite PSQI composite scores, involving sleep quality, sleep latency, sleep duration, sleep efficiency, sleep disturbance, sleep medication use, and daytime dysfunction. Stepwise increases were apparent for five of seven PSQI composite scores (excluding sleep efficiency and sleep medication use) in those with increasing levels of schizotypy (Figure 2).

## 4. Discussion

Unhealthy lifestyle behaviors and chronic health conditions are overrepresented among those diagnosed with schizophrenia. The current study extends this literature by providing evidence for lower endorsement of preventative health behavior and heightened health-related concerns as levels of schizotypy increase in a non-clinical sample. These findings may point towards an earlier origin of physical and behavioral health issues than generally presumed. The differential relationships between symptom subtypes (i.e., positive, negative, disorganized) and health indicators are particularly noteworthy. We have omitted discussion on the ‘psychological health’ LHQ-B variable since the statistically significant findings in the current study can be attributed to construct similarities.

There was a strong positive relationship between levels of schizotypy and somatic symptom scores, with similar patterns emerging across all three schizotypy subscales (Table 2, Table 3 and Table 4). When collapsing across the subscale scores, higher levels of overall schizotypy were significantly related to increased physical symptoms in six of eight categories (Figure 1), including a range of physical complaints including elevated heart rate, upset stomach, muscle ache, migraines, and fatigue. Somatic symptoms can simply represent the bodily manifestation of distress that accompanies a wide range of mental health issues (e.g., depression and anxiety) [64]. However, two well-designed population studies found a significant concurrence of somatic symptoms and psychosis in pre-adolescent and adolescent research participants [16,65]. The relationship between somatic symptoms and psychotic-like experiences appeared to be independent of anxiety or other forms of psychopathology, possibly representing anomalous interpretations of bodily experiences [16]. Of note, there is no current research evaluating the long-term adult outcomes in those with heightened somatic symptoms during early or late adolescence. Similarly, in our cross-sectional sample of young adults, we are unable to determine if the elevated sympathetic (cardiovascular) or metabolic somatic symptoms in those with high levels of schizotypy prospectively relate to cardiometabolic risk factors reported in ultrahigh risk samples [22] or the health and lifestyle issues reported early in the development of a psychotic disorder [13]. Additional research is needed to clarify why these apparent elevations exist and if the assessment of somatic complaints contributes to risk equations for the development of psychosis and future health status.

When examining overall sleep quality (PSQI mean scores), a consistent pattern emerged for all three schizotypy subtypes. In each instance, individuals with high positive, negative, or disorganized symptom scores reported poorer overall sleep quality compared to individuals within the low symptom severity groups (Table 2, Table 3 and Table 4). Post-hoc analyses suggested that increasing levels of schizotypy were related to poorer sleep quality for 5 of 7 PSQI domains (Figure 2). These findings fit well with previous research in at-risk or early psychosis samples [40,42,44]. While the mean scores of our high schizotypy participants did not indicate severe sleep disturbance (i.e., PSQI scores around 9 out of a possible score of 21), they were notably above the severity score commonly used to mark sleep disturbance. The presence of sleep disruption may indicate important clinical changes foreshadowing the emergence of more severe symptoms or behaviors [46,47]. This disruption in sleep may further deplete the cognitive and emotional resources that are generally protecting against the emergence of more severe symptoms and dysfunction in individuals with psychosis liability [43]. In fact, a recent prospective study found that poor sleep quality predicted psychotic-like experiences in those with positive schizotypy [66]. A small scale feasibility study (*n* = 11) reported that the use of a cognitive behavioral intervention in 11 participants with ultra-high risk for psychosis was associated with improved quality of sleep (e.g., mean PSQI baseline scores of 12.5, dropping 5–6 points post-intervention), as well as reduced negative affect and psychotic symptomatology [67]. While still formative, these findings suggest that the sleep quality disturbances associated with the risk for psychosis may represent a target (among others) for preventative efforts.

Other than somatic symptoms, sleep quality, and scores of psychological health, the broader examination of lifestyle behaviors and health indicators yielded a somewhat different picture across the three schizotypy subscales. Scores on health satisfaction (QOLI) were significantly lower in participants with high negative or disorganized schizotypy. These scores reflect the possible presence of health-related discomfort or disability and dissatisfaction with current physical fitness. These ratings may be related to heightened somatic ailments, which were noted across all schizotypy subgroups. Higher negative schizotypy scores were significantly associated with reduced engagement in exercise and poorer perceived fitness relative to peers (i.e., LHQ-B health/exercise) (Table 3). Similarly, there was also a trend for negative symptoms to associate with reduced physical activity in the previous week (i.e., IPAQ). This pattern of relationships appears to be consistent with the clinical literature where negative symptoms are related to reduced physical activity and increased sedentary behaviors [68,69,70,71,72] potentially reflecting low levels of motivation towards physical activity [73]. In this context, the current findings may suggest that negative symptoms and their influence on health behaviors may operate on a continuum that extends into the non-clinical end of the spectrum. If this is the case, it is noteworthy that the negative subscale of the SPQ-BR primarily assesses constricted effect, the number of close friends, and anxiety in social situations [50]. While it is possible that the reduced engagement in exercise and physical activity may reflect a narrowing of the scope of activities involving social interaction, this finding may also represent something occurring in those with high levels of negative schizotypy that is not directly assessed by the SPQ-BR (e.g., low-level manifestations of anhedonia or avolition). This notion receives support given the significant relationship between negative schizotypy and lower scores of *sense of purpose* subscale (Table 3). This scale measures an individual’s endorsement of meaning and connectedness in their own life, which may be related to a reduced interest or engagement in a wide range of activities. Individuals with a low sense of purpose may experience a reduced drive/motivation for demanding activities, similar to what is reported in clinical samples where low motivation is linked to reduced physical activity [73].

Contrary to expectations, no significant differences were indicated between low, intermediate, and high schizotypy groups for any of the substance use categories. There was some indication that alcohol use was *reduced* in the high vs. intermediate and low schizotypy groups. While similar negative relationships between alcohol use and schizotypy have been noted in the literature [74,75], positive relationships have also been reported [76]. These differences may partly reflect differences in how alcohol use is assessed. For example, community-based samples examining participants at-risk for psychosis have reported higher rates of problematic alcohol use despite non-significant differences in daily use rates [25]. Thus, the mixed findings suggest that alcohol use may be highly variable due to assessment procedures and sample characteristics. In the current study, the inverse relationship between alcohol use and levels of negative (interpersonal) schizotypy may be a product of fewer social interactions. There was weak correlational evidence for increased cigarette smoking in those with higher levels of disorganized schizotypy (Table 1). This relationship is consistent with previous research [77]. Increased cigarette smoking in those with greater levels of disorganized schizotypy may be an attempt to improve cognitive difficulties via the short-lived simulating effects of nicotine [78].

There are notable implications when considering the pattern of findings in the current study. First, the increase of somatic symptoms and reduction in health-related wellbeing among those with increased levels of schizotypy suggests that physical health concerns may exist prior to the emergence of psychotic disorder. If this relationship is confirmed by subsequent research, this will challenge the presumption that physical health concerns only emerge as a secondary consequence of antipsychotic medication use or the disruptions in health behaviors (e.g., levels of physical activity) that often accompany clinical symptoms or restricted care environments (e.g., inpatient care; group homes). The current study involved a relatively high-functioning non-clinical sample. Thus the secondary consequences of disorder/treatment (including antipsychotic mediation use) cannot explain the elevated reports of physical health dissatisfaction. However, this does not rule out the possibility that specific schizotypy symptoms may be amplifying physical health complaints (e.g., anomalous interpretations of normative bodily experiences) [16]. Second, the relationship between schizotypy symptom severity, sleep disturbance, and decreased physical activity appear to parallel findings in ultra-high risk and clinical samples. Thus, functional disruptions in these domains, in the context of other risk indicators for psychosis, may be associated with an underlying pathognomonic process. If this is the case, the assessment of these behaviors in screening protocols may improve the accuracy of risk detection. Similarly, if these processes are etiologically relevant, then early detection of sleep disturbance and reductions in physical activity (in the case of negative symptoms) may allow for tailored preventative efforts.

There are several limitations of the current study that should be considered. The participant sample consisted primarily of young adults, largely White, attending a public university in the northeastern region of the United States. Thus, the unique characteristics of this sample limit the generalizability of our findings. In addition, the current project utilized self-report measures which may be associated with response biases or the recall accuracy of health behaviors [79]. Future research should strive to incorporate objective indicators of physical health (e.g., hemoglobin A1C) or lifestyle behaviors (e.g., Fitbit to measure physical activity) to further increase the specificity of the findings. In addition, our data were cross sectional and we were not able to determine if the findings from our study predict greater risk for physical or mental health issues at a later time point. Given these limitations no causal conclusions are possible. Longitudinal research paradigms may help identify if health indicators and behaviors predict future mental and physical health outcomes.

## 5. Conclusions

The current study is among the first to examine a broad range of lifestyle and health indicators in those with varying levels of schizotypy recruited from a non-clinical setting. The overall pattern of findings provides evidence for the presence of problematic lifestyle behaviors and health concerns as the risk for psychosis increases. The findings from the present study are compelling given they generally reflect similar results from ultra-high risk and clinical samples suggesting a continuum. Given the potential prevalence and relevance of these issues, our findings support the expanded inclusion of physical health and lifestyle indicators in high-risk research and prospective studies examining long-term outcomes.

## Figures and Tables

**Figure 1 behavsci-11-00179-f001:**
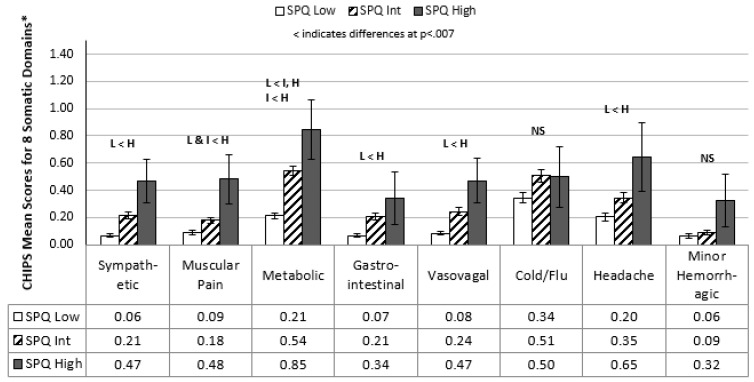
Somatic Symptom Comparisons across Low, Intermediate, and High Schizotypy Scores (Mean/Std. Error). * CHIPS = Cohen-Hoberman Inventory of Physical Symptoms [56] somatic domains [57]. Permission for use of scales is not necessary when use is for nonprofit academic research or nonprofit educational purposes. https://www.cmu.edu/dietrich/psychology/stress-immunity-disease-lab/scales/html/chipsscore.html, accessed on 9 December 2021.

**Figure 2 behavsci-11-00179-f002:**
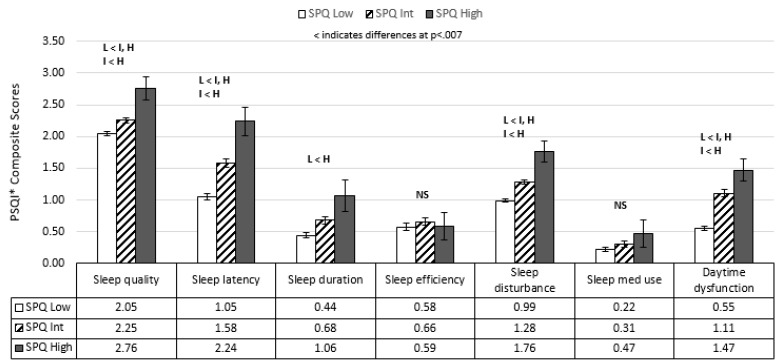
Sleep Quality Comparisons across Low, Intermediate, and High(psychometrically defined) schizotypy categories(Mean/Std. Error). * PSQI = Pittsburgh Sleep Quality Inventory.

**Table 1 behavsci-11-00179-t001:** Bivariate correlations between schizotypy subdomain scores and lifestyle/health variables (*n* = 530).

Measures	1	2	3	4	5	6	7	8	9	10	11	12	13	14	15	16	17	18
1. SPQ-Cognitive-perceptual																		
2. SPQ-Disorganized	0.58 *																	
3. SPQ-Interpersonal	0.53 *	0.53 *																
4. IPAQ-Activity/MET	−0.02	−0.13	−0.17 *															
5. CHIPS-Physical Symptoms	0.39 *	0.34 *	0.26 *	−0.03														
6. Cigarettes (day)	0.12	0.14 *	0.08	−0.08	0.23 *													
7. Alcohol (week)	0.03	0.01	0.03	0.01	0.18 *	0.31 *												
8. Marijuana (week)	0.03	0.04	0.01	0.04	0.15 *	0.33 *	0.38 *											
9. LHQ Health/Exercise	−0.10	−0.18 *	−0.22 *	0.29 *	−0.13	−0.08	0.09	−0.04										
10. LHQ Psychological Health	−0.26 *	−0.23 *	−0.37 *	0.11	−0.35 *	−0.15 *	−0.04	−0.10	0.47 *									
11. LHQ Substance Use	−0.08	−0.01	0.04	0.01	−0.14 *	−0.48 *	−0.57 *	−0.50 *	0.13	0.34 *								
12. LHQ Nutrition	−0.09	−0.10	−0.11	0.17 *	−0.15 *	−0.19 *	−0.13	−0.10	0.41 *	0.33 *	0.34 *							
13. LHQ Environ. Concern	0.06	0.04	0.03	0.05	−0.07	−0.12	−0.13	−0.10	0.25 *	0.37 *	0.37 *	0.51 *						
14. LHQ Social Concern	−0.01	0.06	−0.01	0.00	0.06	−0.10	−0.16 *	−0.15 *	0.29 *	0.45 *	0.46 *	0.27 *	0.48 *					
15. LHQ Prevention	−0.07	0.01	0.02	−0.03	−0.12	−0.18 *	−0.27 *	−0.25 *	0.25 *	0.40 *	0.56 *	0.34 *	0.49 *	0.66 *				
16. LHQ Purpose	−0.03	−0.04	−0.20 *	0.13	−0.08	−0.09	−0.09	−0.10	0.35 *	0.55 *	0.34 *	0.23 *	0.34 *	0.60 *	0.48 *			
17. QOLI Health	−0.16 *	−0.19 *	−0.28 *	0.14 *	−0.29 *	−0.14 *	0.03	−0.09	0.55 *	0.38 *	0.14	0.31	0.18 *	0.20 *	0.19 *	0.31 *		
18. Body Mass Index	0.09	0.03	0.11	0.01	0.03	0.09	0.00	0.01	−0.22 *	−0.16 *	−0.06	−0.06	−0.08	−0.08	−0.06	−0.11	−0.27 *	
19. PSQI Sleep quality	0.36 *	0.33 *	0.37 *	−0.07	0.54 *	0.22 *	0.16 *	0.13	−0.13	−0.43 *	−0.22 *	−0.18 *	−0.10	−0.07	−0.16 *	−0.14 *	0.20 *	0.04

* indicates significance under <0.002. Italicized variables indicate the use of Spearman correlations due to non-normal distributions; SPQ = Schizotypal. Personality Questionnaire-Brief Revised; IPAQ-Activity/MET = International Physical Activity Questionnaire Metabolic Equivalent of Task score; CHIPS = Cohen-Hoberman Inventory of Physical Symptoms; LHQ = Lifestyle and Habits Questionnaire-Brief; QOLI = Quality of Life Inventory; PSQI = Pittsburgh Sleep Quality Inventory.

**Table 2 behavsci-11-00179-t002:** Descriptive information for study variables and MANOVA results organized by high, intermediate, and low Cognitive-Perceptual (positive) schizotypy groups.

	Scores on Cognitive-Perceptual SPQ-BR Subscale										
	Low (*n* = 274)		Intermediate (*n* = 228)		High (*n* = 28)						
Measures	Mean	SD	Mean	SD	Mean	SD	F	Sig	PES ^1^	Power	Post-hoc ^2^
Cognitive-Perceptual (SPQ-BR)	21.23	4.34	35.40	4.65	49.46	4.38	-	-	-	-	NA
Physical Activity-MET (IPAQ)	1496.11	1141.68	1500.60	1111.76	1588.88	1265.85	0.09	0.918	0.00	0.06	
Somatic Symptoms (CHIPS)	11.42	11.65	20.02	16.58	32.64	27.53	37.70	0.000	0.13	1	H > L, I & I > L
Nicotine use (day)	1.85	5.40	2.88	7.00	1.79	4.63	1.87	0.155	0.01	0.39	
Alcohol use (week)	7.00	9.03	7.61	9.43	5.46	8.56	0.80	0.451	0.00	0.19	
Marijuana use (week)	1.51	4.73	2.25	6.47	2.25	5.65	1.16	0.315	0.00	0.25	
Health/Exercise (LHQ-B)	20.22	6.70	19.43	6.28	18.07	6.44	1.92	0.148	0.01	0.40	
Psychological Health (LHQ-B)	26.23	6.02	24.20	4.93	21.25	5.90	15.30	0.000	0.05	0.99	H < L & I < L
Substance Use (LHQ-B)	31.43	8.26	29.97	7.65	30.61	7.76	2.10	0.123	0.01	0.43	
Nutrition (LHQ-B)	11.31	3.78	10.86	3.13	10.07	4.81	2.10	0.123	0.1	0.43	
Environmental Concern (LHQ-B)	15.88	4.27	16.21	3.64	15.89	4.84	0.41	0.662	0.00	0.12	
Social Concern (LHQ-B)	20.90	4.08	20.82	3.16	20.32	4.55	0.31	0.733	0.00	0.01	
Accident Prevention (LHQ-B)	15.88	3.45	15.38	2.77	14.57	2.77	3.16	0.043	0.00	0.61	
Sense of Purpose (LHQ-B)	12.08	2.81	11.81	2.44	11.93	2.61	0.65	0.524	0.00	0.16	
Health Satisfaction (QOLI)	2.05	3.00	1.27	3.02	0.96	3.16	4.98	0.007	0.02	0.81	
Body Mass Index (BMI)	24.21	5.36	24.93	4.79	27.03	8.96	3.98	0.019	0.02	0.71	
Sleep Quality (PSQI)	6.01	2.94	7.63	3.10	9.54	3.30	28.93	0.000	0.10	1	H > L & I > L

^1^ PES = Partial Eta Square; ^2^ Post-hoc comparisons between High (H), Intermediate (I), and Low (Low) schizotypy groupings indicating significance *p* < 0.003; IPAQ-MET = International Physical Activity Questionnaire Metabolic Equivalent of Task score; CHIPS = Cohen-Hoberman Inventory of Physical Symptoms; LHQ-B = Lifestyle and Habits Questionnaire-Brief; QOLI = Quality of Life Inventory; PSQI = Pittsburgh Sleep Quality Inventory.

**Table 3 behavsci-11-00179-t003:** Descriptive information for study variables and MANOVA results organized by high, intermediate, and low Interpersonal (negative) schizotypy groups.

	Scores on Interpersonal SPQ-BR Subscale										
	Low (L) (*n* = 289)		Intermediate (I) (*n* = 213)		High (H) (*n* = 28)						
Measures	Mean	SD	Mean	SD	Mean	SD	F	Sig	PES ^1^	Power	Post-hoc ^2^
Interpersonal (SPQ-BR)	17.42	3.95	29.26	3.76	41.07	2.36	-	-	-	-	NA
Physical Activity-MET (IPAQ)	1622.33	1115.01	1411.25	1137.87	968.23	1112.86	5.51	0.004	0.02	0.85	
Somatic Symptoms (CHIPS)	13.40	14.40	18.88	16.57	25.46	21.89	12.47	0.000	0.05	1.00	H > L & I > L
Nicotine use (day)	2.03	5.79	2.64	6.31	2.29	7.84	0.61	0.544	0.00	0.15	
Alcohol use (week)	7.48	9.33	6.97	9.07	5.75	8.58	0.55	0.578	0.00	0.14	
Marijuana use (week)	1.65	4.99	2.30	6.49	0.86	3.77	1.31	0.271	0.01	0.28	
Health/Exercise (LHQ-B)	20.99	6.33	18.50	6.32	16.64	7.19	12.90	0.000	0.05	1.00	H < L & I < L
Psychological Health (LHQ-B)	26.62	5.84	23.78	4.80	19.32	4.79	33.88	0.000	0.11	1.00	H < L, I & I < L
Substance Use (LHQ-B)	30.70	8.13	30.78	7.76	31.25	8.62	0.06	0.940	0.00	0.06	
Nutrition (LHQ-B)	11.37	3.72	10.73	3.26	10.11	4.33	3.00	0.051	0.01	0.58	
Environmental Concern (LHQ-B)	16.01	4.35	15.98	3.44	16.46	4.92	0.18	0.836	0.00	0.08	
Social Concern (LHQ-B)	21.07	4.10	20.54	3.22	20.61	3.30	1.30	0.272	0.01	0.28	
Accident Prevention (LHQ-B)	15.69	3.35	15.36	2.90	16.43	2.85	1.68	0.187	0.01	0.36	
Sense of Purpose (LHQ-B)	12.39	2.66	11.59	2.45	10.21	2.87	12.47	0.000	0.05	1.00	H < L & I < L
Health Satisfaction (QOLI)	2.36	2.92	0.93	2.97	−0.07	2.83	19.48	0.000	0.07	1.00	H < L & I < L
Body Mass Index (BMI)	24.30	5.17	24.80	5.11	27.48	8.51	4.62	0.010	0.02	0.78	
Sleep Quality (PSQI)	6.06	2.83	7.73	3.33	9.21	2.82	27.07	0.000	0.09	1.00	H > L & I > L

^1^ PES = Partial Eta Square; ^2^ Post-hoc comparisons between High (H), Intermediate (I), and Low (Low) schizotypy groupings indicating significance *p* < 0.003; IPAQ-MET = International Physical Activity Questionnaire Metabolic Equivalent of Task score; CHIPS = Cohen-Hoberman Inventory of Physical Symptoms; LHQ-B = Lifestyle and Habits Questionnaire-Brief; QOLI = Quality of Life Inventory; PSQI = Pittsburgh Sleep Quality Inventory.

**Table 4 behavsci-11-00179-t004:** Descriptive information for study variables and MANOVA results organized by high, intermediate, and low Disorganized schizotypy groups.

	Scores on Disorganized SPQ-BR Subscale										
	Low (*n* = 275)		Intermediate (*n* = 233)		High (*n* = 22)						
Measures	Mean	SD	Mean	SD	Mean	SD	F	Sig	PES ^1^	Power	Post-hoc ^2^
Disorganized (SPQ-BR)	14.15	3.96	25.78	3.33	35.82	1.56	-	-	-	-	NA
Physical Activity-MET (IPAQ)	1620.74	1110.02	1393.09	1167.93	1193.93	902.30	3.43	0.033	0.01	0.64	
Somatic Symptoms (CHIPS)	11.01	10.55	21.12	17.86	29.95	27.03	37.81	0.000	0.13	1.00	H > L & I > L
Nicotine use (day)	1.45	4.92	3.01	6.84	5.09	9.31	6.68	0.001	0.03	0.91	Trend H > L
Alcohol use (week)	7.09	9.22	7.51	9.20	4.86	8.41	0.86	0.424	0.00	0.20	
Marijuana use (week)	1.39	5.00	2.40	6.15	2.27	6.12	2.14	0.118	0.01	0.44	
Health/Exercise (LHQ-B)	20.72	6.63	18.95	6.17	16.45	6.77	7.79	0.000	0.03	0.95	Trend H < L
Psychological Health (LHQ-B)	26.28	6.16	23.99	4.75	21.95	6.11	14.24	0.000	0.05	1.00	H < L & I < L
Substance Use (LHQ-B)	31.06	8.21	30.31	7.75	31.82	7.93	0.75	0.471	0.00	0.18	
Nutrition (LHQ-B)	11.41	3.76	10.61	3.27	11.23	4.25	3.17	0.043	0.01	0.61	
Environmental Concern (LHQ-B)	15.95	4.36	16.10	3.49	16.02	4.04	0.10	0.909	0.00	0.07	
Social Concern (LHQ-B)	20.72	4.20	20.93	2.94	21.23	4.99	0.33	0.720	0.00	0.10	
Accident Prevention (LHQ-B)	15.61	3.48	15.52	2.74	16.23	3.01	0.51	0.601	0.00	0.13	
Sense of Purpose (LHQ-B)	12.17	2.79	11.68	2.42	12.05	2.94	2.21	0.111	0.008	0.45	
Health Satisfaction (QOLI)	2.15	3.05	1.24	2.97	−0.05	2.44	9.60	0.000	0.04	0.98	H < L & I < L
Body Mass Index (BMI)	24.74	5.34	24.38	4.82	26.67	10.18	2.07	0.128	0.01	0.43	
Sleep Quality (PSQI)	6.00	2.82	7.72	3.25	9.36	3.33	27.64	0.000	0.10	1.000	H > L & I > L

^1^ PES = Partial Eta Square; ^2^ Post-hoc comparisons between High (H), Intermediate (I), and Low (Low) schizotypy groupings indicating significance *p* < 0.003; IPAQ-MET = International Physical Activity Questionnaire Metabolic Equivalent of Task score; CHIPS = Cohen-Hoberman Inventory of Physical Symptoms; LHQ-B = Lifestyle and Habits Questionnaire-Brief; QOLI = Quality of Life Inventory; PSQI = Pittsburgh Sleep Quality Inventory.

## Data Availability

Data archived at osf.io/rzmana/ (Schizotypy, lifestyle behaviors, and health indicators). All rights reserved.

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
