# Peer review of "Schizotypy, Lifestyle Behaviors, and Health Indicators in a Young Adult Sample"

_behavsci, 2021, doi:10.3390/bs11120179_

Round 1
Reviewer 1 Report
Dear authors,
Thank you very much for submitting your work for publication in Behavioural Sciences.
This is , indeed, a clinically relevant topic in the psychosis field and, as pointed to in the introduction, has not been properly researched before. I have found the paper to be well-written and the methodology appropriate.
However, I would really appreciate if you could address these minor issues concerning the discussion section prior to its final acceptance.
RESULTS:
First, for the sake of the presentation I would delete all the borders from the tables.
Second, please avoid methodology-related comments (e.g. lines 263-274) which should be included in the methodology section.
DISCUSSION:
First, I think that you could expand the discussion by commenting further on the extent to which these findings may also apply to schizophrenia and related psychoses. In other words, whether there is a continuum from schizoptypy to schizophrenia or they should be considered as separate categories.
Second, one mof the main strengths of this non-clincial sample is the fact that they were not on antipsychotic medication. Hence, the results appear to provide a clearer picture of the relationship between psychosis (by assuming that schizoptypi is part of the psychosis spectrum) and physical health irrespective of treatment. Please, provide some comments on this.
Third, in the discussion section I would avoid statistical terms such as ‘ns’ (line 369).
Fourth, given the cross-sectional design (as highlighted in the limitations section), I would suggest adding that no causality conclusions can be drawn from your analyses, on which you may also provide some further comments. IN addition, although you have stated that “the current project utilized self-report measures which may be associated with response biases or the recall accuracy of health behaviors”, I would really appreciate if you could comment on the extent to which the schizotypy-related assessment may have biased health-related responses and viceversa, and how this may limit the interpretation of the study.
Thank you very much.
Yours sincerely,
Reviewer
Author Response
Thank you for your comments. Please see attached.

Reviewer 2 Report
Thank you for the opportunity to review this interesting manuscript. Overall, this paper has a relevant focus. The research is of interest for researchers and clinicians aiming at improvement of mental healthcare. This is a well-structured and well written paper, and I only have minor remarks. I hope the authors will find the comments helpful for the further process.
Abstract:
I suggest removing the abbreviations from the abstract, or please spell out when introduced for the first time.
Introduction:
This situation with abbreviations is repeated across the manuscript (e.g., UHR - line 72; IV – line 196; CBT – line 333 etc). Please spell out when introduced for the first time.
The Introduction clearly addresses the topic, it is concise, flows well from section to section, and has a clear thesis statement and aims.
Materials and Methods:
Have some inclusion and exclusion criteria been considered por participants’ recruitment? Please clarify.
Line 176- 180. Your symptom categories were based only on cut offs of Cohen’s study? I suggest moving the reference to the beginning or to the end of the sentence. Like it is, in the middle of the sentence is not clear if you consider only these authors.
Results:
Table 1: Please have a closer look to the table, you mentioned that “Italicized variables indicate the use of Spearman correlations due to non-normal distributions;” I could not identify them on the table, perhaps the italic was removed during formatting procedures?
In the same way, **Partial Eta Square (line 191) is not presented in the table.
Discussion:
The discussion section is also well organized and reads very well. There is good use of comparison/contrasting literature. The findings are clear and concise.
Conclusion section: please consider highlighting some clinical implications of your study, this could be an added value for future research.
Author Response

(The authors gave the same response as above.)
